# Comparative efficacy of individually and combined application of compost, biochar, and bentonite on Ni dynamics in a calcareous soil

Hamid Reza Boostani <sup>1\*</sup>, Zahra Jalalpour <sup>2</sup>, Ali Behpouri <sup>2</sup>, Ehsan Bijanzadeh <sup>2</sup>, Mahdi Najafi-Ghiri <sup>1</sup>

- 5 Department of Soil and Water Engineering, College of Agriculture and Natural Resources of Darab, Shiraz
- 6 University, Darab, Iran.
- <sup>2</sup>Department of Agreeocology, College of Agriculture and Natural Resources of Darab, Shiraz University, Darab, Iran.
- 8 Correspondence to: Hamid Reza Boostani (hr.boostani@shirazu.ac.ir)
- 9 **Abstract.** In Iran, Ni-contaminated calcareous soils pose a significant environmental risk, yet 10 effective remediation strategies for these specific conditions remain underexplored. While organic and inorganic amendments are commonly used, their comparative efficacy and potential 11 synergistic effects in combined applications for Ni immobilization are not well-established. To 12 address this, an incubation study investigated the individual and combined effects of municipal 13 solid waste compost (M), its biochar (R), and bentonite (B) on Ni stabilization in a calcareous soil 14 15 at three Ni-contamination levels (0, 150, and 300 mg Ni kg<sup>-1</sup>). Sequential extraction and DTPA-16 release kinetics demonstrated that R was the most effective treatment, significantly reducing labile Ni by transforming it into the residual fraction. This is likely due to its alkaline pH, ash, and 17 phosphorus content, which promote Ni precipitation. In contrast, M increased soil Ni 18 bioavailability. The results revealed that combinations (M+B, R+B, R+M) offered no synergistic 19 advantage. The main finding was that singly-applied municipal solid waste biochar is a superior 20 amendment for Ni immobilization, providing a more efficient and practical remediation strategy 21 22 for contaminated calcareous soils without the need for complex combined treatments.

#### 1 Introduction

27

Heavy metal(loid)s (HMs) are toxic elements that can build up in farmland due to human activities like urbanization, industry, mining, and agrochemical use. Once in the soil, HMs are absorbed by crops, contaminating food and posing serious health risks. They also damage beneficial soil microbes, disrupting nutrient cycles and reducing agricultural output (Jahandari and Abbasnejad, 2024; Faraji et al., 2023; Munir et al., 2021). Nickel (Ni) is a unique HMs that, unlike mercury (Hg), cadmium (Cd), and lead (Pb), is essential for plant growth in minute quantities. However, at concentrations exceeding 35 mg kg<sup>-1</sup> soil, Ni becomes harmful, causing physiological and morphological disruptions that severely impair plant development (Shahzad et al., 2018). In Iran, industrial-adjacent agricultural soils exhibit alarmingly high Ni levels, with an average concentration of 350 mg kg<sup>-1</sup>, far exceeding environmental safety thresholds (Shahbazi et al., 2022). The threshold value for Ni contamination in Iranian soils is 110 mg kg<sup>-1</sup> soil (Shahbazi et al., 2020). A nationwide study analyzing 711 soil samples revealed Ni concentrations ranging from 2.79 to 770 mg kg<sup>-1</sup>, with 11.3% of soils surpassing permissible limits (Shahbazi et al., 2020).

When direct removal of toxic ions is unfeasible, immobilizing them on solid surfaces is the optimal approach (Derakhshan Nejad et al., 2018). To achieve this, materials such as biochar, clay minerals, and bio/polymers are incorporated into soils. These additives, known as "soil modifiers,"

significantly influence the soil's physicochemical properties, sorption capacity, and microbial communities (Jin et al., 2024; Gong et al., 2024; Rasheed and Moghal, 2024). Compost improves HMs immobilization through increased soil organic matter (SOM), which enhances metal complexation and cation exchange capacity (Asemoloye et al., 2020). However, its efficacy varies with decomposition rates and may be limited by potential HMs release during compost mineralization (Hobbs et al., 2011). Biochar, a carbon-rich material produced through biomass pyrolysis, exhibits strong sorption properties due to its large surface area, porous structure, abundance of surface functional groups (e.g., carboxyl, hydroxyl, and phenolic groups), and high aromaticity (Comath et al., 2025; Afshar and Mofatteh, 2024). Its effectiveness in immobilizing HMs in soil depends on factors such as the original feedstock, pyrolysis temperature, particle size, and soil type (Gholizadeh et al., 2024; Fakhar et al., 2025). On the other hand, natural bentonite primarily consists of montmorillonite, a 2:1 clay mineral known for its high cation exchange capacity (CEC) and excellent water retention properties (Zhang et al., 2020). Bentonite clay effectively eliminates HMs from soil, offering a solution due to its permanent surface charges and isomorphic substitution properties (Mi et al., 2020; Xie et al., 2018). Also, from an economic standpoint, bentonite is a highly attractive soil amendment due to its affordability and widespread global availability (Peng and Sun, 2012).

40 41

45

52

64

79

It is widely recognized that the toxicity of HMs is determined by their geochemical fractions rather than their total concentrations in soil (Pelfrene et al., 2020). Additionally, accurately assessing the health risks posed by HMs in contaminated soils relies on how precisely plant uptake is simulated (Krauße et al., 2019). Therefore, integrating sequential extraction methods with HMs release kinetics studies is essential for gaining deeper insights into the behavior of these metals in polluted soils (Wang et al., 2021). Recent studies have commonly employed time-dependent release kinetic experiments using single extractants, along with sequential extraction procedures. to assess how effectively soil amendments immobilize HMs in contaminated soils (Boostani et al., 2024a; Boostani et al., 2023). Analysis of a contaminated soil sample demonstrated that Ni and Cu were predominantly present in residual (Res) fractions (53% and 57%, respectively), with subsequent partitioning in organic (OM), Fe-Mn oxide (FeMnOx), carbonate (Car), and soluble +exchangeable (WsEx) fractions. The application of bentonite significantly altered metal speciation, reducing labile (WsEx) fraction while enhancing retention in FeMnOx and Res phases. This shift effectively decreased the mobility and bioavailability of the HMs, highlighting bentonite's potential as an immobilizing agent in remediation strategies (Gao and Li, 2022a). In another study, Boostani et al. (2023) investigated biochar (cow manure, municipal compost, licorice root pulp; 300/600 °C) effects on Ni stabilization in a Ni-contaminated calcareous soil. They indicated that High-temperature biochars (600 °C) reduced mobile Ni (10-42%) and increased Res Ni (13–38%), with ash-rich biochars most effective.

While previous studies have examined individual organic and inorganic amendments for metal immobilization, a critical knowledge gap exists regarding the comparative and synergistic efficacy of compost, biochar, and bentonite on Ni immobilization in contaminated calcareous soils. It remains uncertain whether combined applications offer a superior strategy over single amendments for immobilizing Ni in calcareous environments, a key question this study aimed to answer. Based on the well-documented properties of high stability, alkaline pH, and significant surface area in

biochars, we hypothesized that biochar would be more effective than compost or bentonite at immobilizing Ni, by converting more labile fractions into the stable residual form. Furthermore, given the distinct mechanisms of metal retention offered by organic (e.g., complexation) and inorganic (e.g., adsorption, ion exchange) amendments, we also hypothesized that their combined application would exhibit a synergistic effect, leading to a greater reduction in Ni bioavailability and desorption than any amendment applied alone. Therefore, the goals of the present study were to (i) Compare the effectiveness of municipal solid waste compost, its derived biochar, and bentonite, both individually and in combination, on immobilizing Ni in an artificially contaminated calcareous soil by assessing changes in its chemical fractions, (ii) Determine the kinetics of Ni release in the amended soils to evaluate the stability of immobilized Ni, and (iii) Identify the primary mechanisms responsible for Ni immobilization by employing advanced analytical techniques.

# 2 Materials and methods

## 2.1 Soil Collection, Characterization, and Ni Contamination

A topsoil sample (0-30 cm) was collected from a calcareous soil in Darab, southern Iran, using an auger. The soil was air-dried immediately, sieved through a 2 mm mesh, and stored at room temperature for further physical and chemical analyses. Standard laboratory methods (Sparks et al., 2020) were employed for soil characterization. For Ni contamination, 2 kg of soil was placed in plastic containers and mixed with 500 mL of a NiCl<sub>2</sub> solution to achieve a final concentration of 150 and 300 mg Ni kg<sup>-1</sup> soil. The treated soil was then air-dried at ambient temperature, brought to field capacity (FC) using distilled water, and dried again. This wetting-drying cycle was repeated three times to ensure Ni equilibration, simulating natural field conditions (Boostani et al., 2024a). The selection of a calcareous soil for this study was motivated by its prevalence in agricultural soils of Iran and the need to address a knowledge gap regarding Ni dynamics in high-pH environments. Contrary to the well-established principle that high pH reduces metal solubility, recent studies suggest that specific pedogenic factors and anthropogenic contamination may facilitate increased Ni mobilization in calcareous soils (Shahbazi et al., 2022).

## 2.2 Biochar preparation and its characteristics

Municipal solid waste compost was chosen as feedstock for biochar production. It was first airdried and finely ground. The processed biomass was then oven-dried at 105 °C for 24 hours. Slow pyrolysis was carried out in an electric muffle furnace (Shimifan, F47) under limited oxygen conditions (Khalili et al., 2024). The temperature was gradually increased at a rate of 5 °C per minute from room temperature until reaching 500 °C, which was maintained for two hours. After pyrolysis, the biochars were allowed to cool naturally and then sieved through a 0.5 mm mesh to ensure consistent particle size. The chemical properties of the biochars were assessed using standard laboratory methods (Singh et al., 2017). Fourier Transform Infrared (FTIR) spectroscopy (Shimadzu DR-8001) with the KBr pellet technique was used to analyze surface functional groups. Additionally, the surface morphology of the biochars was examined using scanning electron microscopy—Energy Dispersive X-ray Spectroscopy (SEM-EDX) (TESCAN-Vega3, Czech Republic). X-ray diffraction (XRD) analysis was performed using a GNR XRD Explorer diffractometer (Italy) with Cu Kα radiation (λ = 1.54178 Å), scanning a 2θ range of 10° to 70°.

## 2.3 Preparation and composition of bentonite

The bentonite used in this research was sourced from natural mines in Semnan province, Iran. The raw material was crushed and sieved to achieve a particle size below 0.5 mm. Table 1 presents the elemental analysis of the bentonite powder employed in the study.

**Table 1.** Elemental composition of the natural bentonite (%).

| Cl   | TiO <sub>2</sub> | MgO  | CaO  | K <sub>2</sub> O | Na <sub>2</sub> O | $Fe_2O_3$ | $Al_2O_3$ | $SiO_2$ |
|------|------------------|------|------|------------------|-------------------|-----------|-----------|---------|
| 0.48 | 0.2              | 2.68 | 0.39 | 0.49             | 2.11              | 1.64      | 11.93     | 70.56   |

# 2.4 Experimental Design and Incubation Procedure

The experiment followed a factorial arrangement in a completely randomized design with three replicates. The treatments consisted of three Ni levels (0 (Ni<sub>0</sub>), 150 (Ni<sub>1</sub>), and 300 (Ni<sub>2</sub>) mg kg<sup>-1</sup> soil supplied as NiCl<sub>2</sub>) and Seven soil amendment types: control (no amendment, Cl), municipal solid waste compost (M), bentonite (B), municipal solid waste compost biochar (R), M+B, R+B and R+M). Each amendment was applied at a 2% (w/w) rate (Jiang et al., 2024). 200 g of Contaminated soil samples (spiked with 150 and 300 mg Ni kg<sup>-1</sup>) was placed in plastic containers and thoroughly mixed with the designated amendments. The soil moisture was adjusted to near field capacity using distilled water. The samples were incubated at 25±2°C for 90 days, with moisture levels maintained by daily additions of distilled water. After incubation, the soil was airdried, sieved (2 mm), and stored for subsequent chemical analysis.

# 2.5 Fractionation of soil Ni

Nickel in the soil was partitioned into five chemical fractions using the sequential extraction procedure described by Salbu and Krekling (1998). Detailed steps and reagents used for the sequential extraction procedure was given in the Table 2. The fractions included: water-soluble + exchangeable (WsEx), carbonate-bound (Car), iron-manganese oxide-bound (FeMnOx), organic matter-bound (OM) and residual (Res). The residual fraction was determined by subtracting the sum of the first four fractions from the total soil Ni content. Total Ni concentration in the soil was measured following the method of Sposito et al. (1982). Briefly, the total Ni concentration was quantified via acid digestion of 2 g of soil using 12.5 mL of 4 N HNO<sub>3</sub> at 80 °C for approximately 12 hours, followed by analysis of the resulting extract.

Table 2. Sequential extraction procedure (Salbu and Krekling, 1998).

| Fractions                   | Reagent                                                                                 | Agitation time                |  |
|-----------------------------|-----------------------------------------------------------------------------------------|-------------------------------|--|
| Soluble+exchangeable (WsEx) | 20 mL of 1 M NH <sub>4</sub> OAc (pH=7)                                                 | 2 h                           |  |
| Carbonate-bound (Car)       | 20 mL of 1 M NH <sub>4</sub> OAc (pH=5)                                                 | 2 h                           |  |
| FeMnOxide-bound (FeMnOx)    | 20 mL of 0.04 M NH <sub>2</sub> OH. HCl in 25 % (v/v)<br>CH <sub>3</sub> COOH           | 6 h at 60°C                   |  |
| Organic-bound (OM)          | 15 mL of 30 % (v/v) $H_2O_2$ (pH=2, HNO <sub>3</sub> )                                  | 5.5 h at 80°C (on water bath) |  |
|                             | and after cooling, 5 mL of 3.2 M NH <sub>4</sub> OAc in 20 % ( $v/v$ ) HNO <sub>3</sub> | 30 min                        |  |
| Residual-bound (Res)        | Calculated by subtracting the sum of four fractions from total Pb                       |                               |  |

# 2.6 Nickel desorption study

- The Ni desorption experiment was conducted by mixing 10 g of air-dried soil with 20 mL of
- DTPA solution (pH 7.3) (Lindsay and Norvell, 1978). The extraction process was performed using
- an end-over-end shaker at  $25 \pm 2^{\circ}$ C for varying time intervals (5, 15, 30, 60, 120, 360, 720, and
- 1440 minutes). Following each extraction period, samples were centrifuged at 4000 rpm for 10
- minutes. The resulting supernatants were filtered through Whatman No. 42 filter paper, and Ni
- concentrations were determined by atomic absorption spectroscopy (AAS; PG 990, PG
- Instruments Ltd., UK). The time-dependent Ni release (q<sub>t</sub>) was modeled using a power function
- kinetic equation ( $q_t = at^b$ ). Model validity was assessed based on high coefficient of determination
- (r<sup>2</sup>) values and low standard error of estimate (SEE).

# 160 2.7 Statistical Analysis

- All data were processed using MSTATC software. Treatment means were compared using
- Tukey's multiple range test at  $p \le 0.05$ . Graphical representations were prepared in Microsoft
- Excel 2013, while Pearson correlation analyses were performed with SPSS 22.0 software.
- 2.8 Quality assurance/quality control (QA/QC) procedures
- 2.8.1 *Soil and amendment preparation*
- All soils and amendments (R, M, B) were processed using clean, acid-washed tools to prevent
- cross-contamination. Samples were homogenized by thorough mixing before subsampling for
- analyses.
- 2.8.2 *Replicates and blanks*
- All treatments were performed in triplicate, and reagent blanks were included during sequential
- extraction and desorption procedures to monitor potential contamination.
- 2.8.3 Calibration and standards
- Nickel concentrations were quantified using AAS (PG 990, PG Instruments Ltd., UK). The
- instrument was calibrated daily with a series of standard Ni solutions prepared from certified stock
- solutions (Merck, Germany). A calibration curve with  $r^2 > 0.99$  was obtained before each run.
- Quality control standards were analyzed after every 10 samples to confirm instrument stability.
- 2.8.4 Detection limits and precision
- The detection limit for Ni with the PG990 in flame mode was ~3 μg L<sup>-1</sup> under optimized
- conditions. Based on this detection limit and our extraction protocols, which used a Y g soil sample
- diluted to a final volume of Z mL, the corresponding method detection limit for Ni in soil was
- calculated as mg kg<sup>-1</sup>. Analytical precision was verified by duplicate sample runs, with relative
- standard deviations (RSDs) consistently below 5%.
- 2.8.5 Sequential extraction

To minimize redistribution or re-adsorption of Ni during sequential extraction, all extractions were performed in acid-cleaned polypropylene centrifuge tubes with constant agitation under controlled conditions.

## 2.8.6 Statistical validation

Normality and homoscedasticity were confirmed to validate the assumptions for ANOVA and Tukey's test.

# 3 Results and discussions

#### 3.1 Soil characteristics

Table 3 summarizes the pre-contamination physicochemical properties of the soil, characterized by a sandy loam textural class. The soil was slightly alkaline, with elevated calcium carbonate (42.0%), indicating potential phosphorus fixation and reduced micronutrient availability. Low electrical conductivity (0.58 dS m<sup>-1</sup>) confirms negligible salinity, while minimal organic carbon (0.39%) suggests restricted nutrient cycling. These attributes are consistent with the soil characteristics found in the arid and semi-arid regions of Iran (Mirkhani et al., 2010). Although total Ni (38.0 mg kg<sup>-1</sup>) was slightly higher than the typical background level (35 mg kg<sup>-1</sup>) (Shahzad et al., 2018), its low DTPA-extractable fraction (0.20 mg kg<sup>-1</sup>) indicates limited bioavailability under prevailing conditions.

**Table 3.** Some physicochemical characteristics of the soil sample prior to contamination.

|                                                                | •           |
|----------------------------------------------------------------|-------------|
| characteristics                                                | value       |
| Sand (%)                                                       | 53.7 (1.20) |
| Silt (%)                                                       | 24.3 (2.10) |
| Clay (%)                                                       | 22.0 (1.30) |
| Soil textural class                                            | Sandy loam  |
| $pH_s$                                                         | 7.50 (0.01) |
| Electrical conductivity (dS m <sup>-1</sup> )                  | 0.58 (0.01) |
| Soil organic carbon (%)                                        | 0.39 (0.03) |
| Cation exchange capacity (cmol <sub>c</sub> kg <sup>-1</sup> ) | 12.0 (1.18) |
| Calcium carbonate equivalent (%)                               | 42.0 (2.80) |
| Ni-DTPA (mg kg <sup>-1</sup> )                                 | 0.20 (0.01) |
| Total Ni (mg kg <sup>-1</sup> )                                | 38.0 (1.90) |

### 

# 3.2 Chemical properties of the soil amendments

Table 4 shows the chemical properties of three soil amendments with varying physicochemical properties. The R had the highest electrical conductivity (2.26 dS m<sup>-1</sup>) and pH (9.12) among the amendments, followed by M (1.84 dS m<sup>-1</sup>, 7.42) and B (0.83 dS m<sup>-1</sup>, 8.40). Organic-rich amendments (R and M) contain quantifiable amounts of carbon (23.0%, 18.1%), nitrogen (1.85%, 2.50%), and phosphorus (0.42%, 0.29%), but B does not contain these nutrients. The R is also significantly greater in ash content (65.0%) and Ni concentration (21.6 mg kg<sup>-1</sup>) compared to the M (50.0%, 14.0 mg kg<sup>-1</sup>), reflecting its pyrolyzed nature. The H:C mole ratio, a measure of aromaticity (Mccall et al., 2024), is lower in the R than in M, reflecting greater carbon stability in

Table 4. Chemical characteristics of applied amendments.

| Table 4. Chemical characteristics of applied amendments. |             |             |             |  |  |  |  |
|----------------------------------------------------------|-------------|-------------|-------------|--|--|--|--|
| Characteristics                                          | M           | R           | В           |  |  |  |  |
| EC (1:20) (dS m <sup>-1</sup> )                          | 1.84 (0.02) | 2.26 (0.03) | 0.83 (0.01) |  |  |  |  |
| pH (1:20)                                                | 7.42 (0.01) | 9.12 (0.04) | 8.40 (0.01) |  |  |  |  |
| Total C (%)                                              | 18.1 (0.35) | 23.0 (0.41) |             |  |  |  |  |
| Total H (%)                                              | 3.40 (0.04) | 2.44 (0.03) |             |  |  |  |  |
| Total N (%)                                              | 2.50 (0.03) | 1.85 (0.02) |             |  |  |  |  |
| Total P (%)                                              | 0.29 (0.10) | 0.42 (0.13) |             |  |  |  |  |
| Total Ni (mg kg <sup>-1</sup> )                          | 14.0 (1.08) | 21.6 (2.10) | Nd          |  |  |  |  |
| Ash Content (%)                                          | 50.0 (2.03) | 65.0 (2.60) |             |  |  |  |  |
| *O+S content (%)                                         | 23.0 (0.28) | 7.71 (0.21) |             |  |  |  |  |
| H:C mole ratio                                           | 2.25 (0.02) | 1.27 (0.01) |             |  |  |  |  |

**Notes:** M, Municipal solid waste compost; R, Municipal solid waste compost biochar; B, Bentonite clay; \*Determined by subtraction of ash, moisture, C, N and H from total mass

## 3.3 FTIR, SEM-EDX and XRD of the soil amendments

FTIR spectrum of R, produced through pyrolysis of compost, exhibits several changes as a result of thermal degradation during biochar production (Figure 1). Most notably, decreased intensity or absence of C=O stretching vibrations near 1700 cm<sup>-1</sup> (carboxyl groups) (Keiluweit et al., 2010) in the R compared to M suggests a loss in CEC, which is corroborated by the lower O+S content of R (Table 4). The intense band at 1432 cm<sup>-1</sup> in R may suggest greater contents of CaCO<sub>3</sub> or lignin-derived structures (Myszka et al., 2019) compared to M. Conversely, the weakened band at 1100 cm<sup>-1</sup> specific for cellulose (Keiluweit et al., 2010) in R suggests thermal degradation of the labile organic constituents by pyrolysis, once again pointing toward the structural transformation from compost to biochar. The FTIR spectrum of R also shows characteristic mineral-associated vibrations, with Si-H stretching bands at 470 cm<sup>-1</sup> and 800 cm<sup>-1</sup> easily recognizable (Zemnukhova et al., 2015). The spectral features likely indicate the formation of organo-mineral complexes as a result of thermal transformation processes during pyrolysis (Lehmann and Joseph, 2024).

**Figure 1.** FTIR spectra for the organic amendments (M and R) in the range of 400-2000 cm<sup>-1</sup> Notes: M, Municipal solid waste compost; R, Municipal solid waste compost biochar

The EDX analysis showed distinct elemental composition for each amendment, in agreement with their respective organic or inorganic sources (Figure 2). Bentonite clay contained predominantly silicon (Si, 33.9%) and aluminum (Al, 7.10%), in agreement with its aluminosilicate mineral nature. Trace levels of elements such as iron (Fe, 1.99%) and alkali metals (Na and K) reflect the presence of natural impurities. On the other hand, the organic amendments contained significantly higher carbon (C, 19.9% for M; 26.1% for R) and nitrogen (N, 4.22% for M; 3.50% for R) content, which is a reflection of their compost-derived organic matter. R had higher carbon content due to higher carbonization from the pyrolysis process. In addition, R contained a significantly higher proportion of calcium (Ca, 22.2%), possibly originating from mineral additives or ash materials developed during the production of biochar.

The SEM images also presented surface structural differences in the amendments (Figure 2). Bentonite clay showed a close, laminated structure with level, platelet-shaped particles characteristic of aluminosilicate minerals, in line with its high Si and Al percentages. The M possessed a heterogeneous, porous structure with irregularly sized organic pieces, consistent with its organic origin and partially decomposed biomass. In contrast, the R possessed an extremely porous, broken morphology with a honeycomb structure, the outcome of pyrolytic treatment that maximizes surface area and promotes carbon retention.

**Figure 2.** SEM-EDX of the organic and inorganic amendments (M, R and B). Notes: M, Municipal solid waste compost; R, Municipal solid waste compost biochar; B, Bentonite clay.

The XRD analysis of the soil amendments contained distinct crystalline phases: the M exhibited a broad hump  $(2\theta \sim 20-35^\circ)$  typical for amorphous organic matter (e.g., cellulose, lignin) (Noorshamsiana et al., 2020) with quartz peaks  $(2\theta \sim 26.6^\circ)$  (Zuo et al., 2016) and minor traces of

calcite (minor peaks at ~40-50°) (Al-Jaroudi et al., 2007). The R showed sharp graphitic carbon peaks  $(2\theta \sim 50^\circ)$  (Destyorini et al., 2021) due to pyrolysis-induced crystallinity, along with persistent quartz  $(2\theta \sim 26.6^\circ)$  and potential calcium oxalate or phosphate (Peaks near 30°) phases (Petrova et al., 2019). Bentonite clay was dominated by montmorillonite reflections (20: 19.8°, 35°, and 62°), with quartz impurities  $(2\theta: 21.8^\circ, 26.6^\circ, 35.9^\circ)$  (Ikhtiyarova et al., 2012).

269270

264265

**Figure 3.** The XRD patterns for all the soil amendments (M, R and B). Notes: M, Municipal solid waste compost; R, Municipal solid waste compost biochar; B, Bentonite clay.

# 3.4 Changing the soil pH as affected by different amendments

The influence of Ni levels, soil amendments, and their interactions on soil pH was statistically significant (P < 0.01). Elevating Ni levels from Ni<sub>0</sub> to Ni<sub>2</sub> resulted in a 0.03 unit decrease in soil pH, whereas no significant difference was observed between Ni<sub>0</sub> and Ni<sub>1</sub> levels (Table 5). This pH reduction at higher Ni concentrations may be attributed to Ni-induced displacement of exchangeable H<sup>+</sup> ions or the hydrolysis of Ni<sup>+2</sup>, which releases protons into the soil solution (Sparks et al., 2022). Consistent with the present study, elevating Ni application from 150 to 300 mg Ni kg<sup>-1</sup> in a calcareous soil contaminated with Ni resulted in an average reduction of 0.20 units in soil pH (Boostani et al., 2020). In contrast, amendments differentially increased soil pH, with the most pronounced effect (0.24 units) recorded for the R+B treatment, while the minimal increase (0.05 units) was associated with M (Table 5). The pH-enhancing effect of amendments (M, R, and B) can be ascribed to their liming properties, as confirmed by the FTIR (Figure 1), EDX (Figure 2) and chemical composition (Table 4) analysis. Both biochar and bentonite exhibit inherent alkalinity, consistent with their established role in mitigating soil acidity (Xie et al., 2024). The increase in soil pH observed with the B addition could be attributed to its high levels of SiO<sub>2</sub>. MgO, and CaO (Table 1), which help neutralize soil acidity (Afzal et al., 2024). Additionally, biochar contributes to higher pH by exchanging its surface salt-based ions (Na<sup>+</sup>, Ca<sup>2+</sup>, K<sup>+</sup>, and

Mg<sup>2+</sup>) with acidic H<sup>+</sup> in the soil, thereby boosting salt-based ion saturation and improving soil alkalinity (Lin et al., 2023). Notably, combined amendments like R+B, M+B yielded higher soil pH values (7.61 and 7.57, respectively) than individual applications (M, B, R) (Table 4), suggesting synergistic mechanisms wherein bentonite's high CEC may enhance retention of base cations derived from biochar or compost, thereby further neutralizing soil acidity. In agreement with our results, Xie et al. (2024) observed that a 1:1 weight ratio mixture of corn straw biochar and bentonite led to a greater rise in soil pH compared to their separate applications. Interaction analysis revealed the highest and lowest pH values in the Ni<sub>0</sub>×R+B (7.69) and Ni<sub>0</sub>×Cl (7.21) treatments, respectively. Additionally, Ni application appeared to attenuate soil pH variability across amendments compared to the Ni<sub>0</sub> level (Table 5).

**Table 5.** Changing the soil pH as affected by the application of different amendments

|              | Cl                | M                 | В                  | R                  | M+B                | R+B                | R+M               | Main effects      |
|--------------|-------------------|-------------------|--------------------|--------------------|--------------------|--------------------|-------------------|-------------------|
| Nio          | 7.21 <sup>j</sup> | 7.36 <sup>i</sup> | 7.53 <sup>de</sup> | 7.47 <sup>fg</sup> | 7.63 b             | 7.69 a             | 7.60 bc           | 7.50 <sup>A</sup> |
| 1810         | (0.020)           | (0.015)           | (0.025)            | (0.005)            | (0.045)            | (0.015)            | (0.010)           | 7.50              |
| Niı          | 7.49 e-g          | $7.47^{fg}$       | 7.54 <sup>de</sup> | $7.47^{fg}$        | 7.57 <sup>cd</sup> | 7.54 <sup>de</sup> | $7.48^{fg}$       | 7.51 <sup>A</sup> |
| INII         | (0.015)           | (0.005)           | (0.010)            | (0.005)            | (0.005)            | (0.005)            | (0.010)           |                   |
| $Ni_2$       | 7.41 <sup>h</sup> | 7.42 h            | $7.47^{\rm fg}$    | 7.42 h             | 7.51 <sup>ef</sup> | $7.60^{\ bc}$      | 7.45 gh           | 7.47 <sup>B</sup> |
|              | (0.005)           | (0.005)           | (0.025)            | (0.005)            | (0.005)            | (0.005)            | (0.005)           | 7.47              |
| Main effects | 7.37 <sup>F</sup> | $7.42^{\rm E}$    | 7.51 <sup>C</sup>  | 7.45 <sup>D</sup>  | 7.57 <sup>B</sup>  | 7.61 <sup>A</sup>  | 7.51 <sup>C</sup> |                   |

**Notes:** Cl, control (without amendment addition); M, municipal solid waste compost; B, bentonite clay; R, municipal solid waste compost biochar;  $Ni_0$ , without  $Ni_0$  application;  $Ni_1$ , application of 150 mg  $Ni_0$  kg<sup>-1</sup> soil;  $Ni_0$ , application of 300 mg  $Ni_0$  kg<sup>-1</sup> soil. Numbers with same lower letters (interaction effects), in each section, had no significance difference to each other (P < 0.05). Also, numbers with same capital letters (main effects), in each section, had no significance difference to each other (P < 0.05).

# 3.5 Changes of soil Ni chemical fractions as affected by the amendment application

The Ni levels, soil amendment types, and their interactions exhibited statistically significant influences on Ni concentrations in the WsEx, Car and Res fractions (Table 6). Overall, elevating Ni levels from Ni<sub>0</sub> to Ni<sub>2</sub> resulted in significant increases in Ni concentrations across all chemical fractions, with the most pronounced rise observed in the FeMnOx and OM fractions (16.8-fold and 15.4-fold), respectively. In calcareous soils, the elevated pH (typically >7.5) enhances the formation of FeMn oxides/hydroxides with numerous reactive sites, which act as strong sinks for Ni via specific adsorption, co-precipitation, and surface complexation (Sparks et al., 2022; Rinklebe et al., 2016). Furthermore, Ni strongly complexes with organic ligands, particularly humic and fulvic acids, via carboxyl and phenolic functional groups, leading to its preferential retention in organic matter-bound fractions under aerobic conditions (Huang et al., 2023). The present findings align with prior research by Boostani et al. (2020), which identified Res, OM and FeMnOx fractions as the dominant chemical forms of Ni in a highly Nicontaminated calcareous soil. Moreover, Mailakeba and Bk (2021) demonstrated that increasing the application rate of Ni (supplied as NiCl<sub>2</sub>) from 0 to 180 mg kg<sup>-1</sup> soil led to a substantial elevation in the soil Ni content within the OM fraction.

The efficacy of soil amendments in altering Ni concentrations within the WsEx fraction varied depending on initial soil Ni levels (Table 6). In the absence of added Ni (Ni<sub>0</sub>), all

amendments except M significantly reduced Ni-WsEx concentrations relative to the CL, with the most substantial decrease (60.7%) observed in the R treatment (Table 6). Conversely, at the Ni<sub>1</sub> level, R, R+B, and R+M treatments reduced Ni-WsEx concentrations, though only R demonstrated statistically significant effect (Table 6). In contrast, at the Ni<sub>2</sub> level, amendments containing M (M, M+B, and R+M) significantly increased Ni-WsEx content compared to the CL, whereas other amendments had no significant impact (Table 6). Overall, the application of M consistently led to an increase in Ni concentrations within the WsEx fraction across all the Ni levels (Table 6). Additionally, the effectiveness of combined amendments containing M, such as M+B and R+M, in reducing Ni concentrations in the WsEx fraction appeared to be lower compared to the application of each component individually (B and R) (Table 6). The M possesses a limited number of binding sites; however, its capacity to retain HMs is influenced by both the metal's affinity for the binding sites and the M's pre-existing trace element load. While Pb exhibits strong binding affinity, Ni does not (Cao et al., 2023). The M contained considerable concentrations of Ni (Table 4) as well as other competing cations such as Ca, Mg, and K (Figure 2). Consequently, the available binding sites may become saturated, thereby reducing the M's effectiveness in immobilizing newly introduced Ni. In other point of view, during the microbial decomposition process, M may release dissolved organic carbon (DOC), which can chelate Ni and maintain it in soluble forms (Flury et al., 2015). Conversely, among all treatments, R demonstrated the greatest efficacy in reducing Ni concentrations in the WsEx fraction under non-contaminated (Ni<sub>0</sub>) and moderately contaminated (Ni<sub>1</sub>) conditions. This result can likely be attributed to the alkaline properties of R, characterized by its high pH, ash content, and phosphorus levels (Table 4), which promote the precipitation of Ni as hydroxides, carbonates, and phosphates in the soil (Boostani et al., 2024b). Nevertheless, at the highest contamination level (Ni<sub>2</sub>), the application of R did not significantly affect soil Ni content in the WsEx fraction (Table 6). Although the porous structure of R (Figure 2) provides numerous adsorption sites for Ni<sup>+2</sup> ions, these sites may become saturated under high Ni concentrations, resulting in the persistence of residual Ni within the WsEx fraction (Liang et al., 2021).

321

325

331332

357

360

The experimental soil, despite its calcareous nature, exhibited unexpectedly low Ni concentrations in the Car fraction compared to the Res and FeMnOx fractions (Table 5 and Figure 4). This observation suggests that either the standard 1 M sodium acetate (pH 5) extraction method may be insufficiently aggressive to fully dissolve carbonate-bound Ni, or that Ni demonstrates only marginal chemical preference for carbonate association in such soil environments (Rajaie et al., 2008). Statistical analysis of interactions revealed that under Ni<sub>0</sub> conditions, none of the amendments, whether applied singly or in combination, significantly altered the Ni content in the Car fraction (Table 6). At moderate Ni contamination (Ni<sub>1</sub>), only R, B, and their combined application (R+B) produced a statistically significant reduction in Ni concentration within the Car fraction relative to the control, with no notable differences observed among these treatments (Table 6). Boostani et al. (2020) demonstrated that incorporating various crop residue-derived biochars into a Ni-contaminated calcareous soil effectively decreased Ni concentration within the Car fraction. Furthermore, previous research supports the observations regarding the bentonite application, with Gao and Li (2022b) documenting dose-dependent reductions in Ni content in the Car fraction following bentonite application to contaminated soils. Parallel findings by Boostani et al. (2025) stablished an

inverse relationship between bentonite application rates and Cd concentration in the Car fraction of a polluted calcareous soil. Conversely, under high Ni contamination (Ni<sub>2</sub>), treatments involving M alone or in combination with B and R (M+B, M+R) led to a marked elevation in the Car fraction compared to the control. The most substantial increase (50.8%) was recorded for the sole application of M (Table 6). In contrast, R, B, and their combined use (R+B) exhibited no significant influence on Ni content in this fraction at the Ni<sub>2</sub> level (Table 6).

**Table 6.** Soil Ni content in fractions of WsEx, Car and Res as affected by the application of amendments

|                 | Cl                   | M                   | В                    | R                    | M+B                | R+B                  | R+M                | Main effects       |
|-----------------|----------------------|---------------------|----------------------|----------------------|--------------------|----------------------|--------------------|--------------------|
| •               |                      |                     |                      | WsEx-Ni              |                    |                      |                    |                    |
| NT:             | 6.75 e-g             | 7.15 e-g            | 4.30 h-j             | 2.65 <sup>j</sup>    | 4.15 h-j           | 3.57 <sup>ij</sup>   | 3.65 <sup>ij</sup> | 4.62 C             |
| $Ni_0$          | (0.05)               | (0.25)              | (0.10)               | (0.15)               | (1.45)             | (1.75)               | (0.75)             | 4.62 <sup>C</sup>  |
| NT:             | 8.09 d-f             | 9.40 <sup>cd</sup>  | 8.32 d-e             | 5.69 g-i             | 10.50 bc           | 6.15 f-h             | 7.79 d-f           | 7 00 B             |
| Ni <sub>1</sub> | (0.50)               | (0.48)              | (0.22)               | (0.38)               | (0.08)             | (0.35)               | (0.53)             | 7.99 <sup>B</sup>  |
| NT2             | 12.20 b              | 17.19 a             | 12.29 b              | 10.53 bc             | 16.57 a            | 10.53 bc             | 15.54 <sup>a</sup> | 12 55 A            |
| Ni <sub>2</sub> | (0.60)               | (0.25)              | (0.26)               | (0.08)               | (1.23)             | (0.22)               | (0.45)             | 13.55 A            |
| Main effects    | 9.01 <sup>A</sup>    | 11.25 <sup>A</sup>  | 8.30 <sup>B</sup>    | 6.29 <sup>C</sup>    | 10.40 A            | 6.81 <sup>C</sup>    | 8.99 <sup>B</sup>  |                    |
| •               |                      |                     |                      | Car-Ni               |                    |                      |                    |                    |
| <b>3</b> . ₹•   | 4.75 h               | 3.89 h              | 3.69 h               | 3.77 h               | 3.83 h             | 3.82 h               | 3.79 h             | 3.93 <sup>C</sup>  |
| $Ni_0$          | (0.25)               | (0.08)              | (0.10)               | (0.05)               | (0.14)             | (0.10)               | (0.02)             |                    |
| NT:             | 18.51 <sup>f</sup>   | 19.25 ef            | 14.08 <sup>g</sup>   | 15.24 <sup>g</sup>   | 20.67 d-f          | 15.73 <sup>g</sup>   | 19.64 ef           | 17.59 <sup>B</sup> |
| Ni <sub>1</sub> | (0.48)               | (0.06)              | (0.50)               | (0.26)               | (0.31)             | (0.05)               | (1.58)             |                    |
| NT:             | 21.28 <sup>c-e</sup> | 32.09 a             | 22.98°               | 22.89 cd             | 29.34 b            | 21.49 <sup>c-e</sup> | 28.80 b            | 25.55.4            |
| Ni <sub>2</sub> | (0.71)               | (0.66)              | (1.29)               | (0.83)               | (1.26)             | (0.77)               | (1.50)             | 25.55 A            |
| Main effects    | 14.85 B              | 18.41 A             | 13.58 <sup>°C</sup>  | 13.97 <sup>°</sup> C | 17.94 A            | 13.68 <sup>°</sup> C | 17.41 A            |                    |
| •               |                      |                     |                      | Res-Ni               |                    |                      |                    |                    |
| <b>3</b> .70    | 15.81 <sup>m</sup>   | 31.101              | 20.34 m              | 42.76 k              | 33.86 kl           | 41.88 k              | 55.71 <sup>j</sup> | 24.40.0            |
| $Ni_0$          | (0.38)               | (0.09)              | (0.13)               | (0.01)               | (1.23)             | (1.18)               | (1.62)             | 34.49 <sup>C</sup> |
| 270             | 76.70 <sup>° i</sup> | 79.63               | 93.27 eh             | 100.6 fg             | 86.65 hi           | 102.6 fg             | 104.8 <sup>f</sup> | 02 01 B            |
| Ni <sub>1</sub> | (2.96)               | (1.76)              | (1.60)               | (5.17)               | (0.72)             | (1.25)               | (3.30)             | 92.01 <sup>B</sup> |
| NT:             | 132.8 <sup>'d</sup>  | 118.6 <sup>e</sup>  | 146.1 <sup>°</sup> c | 172.1 <sup>a</sup>   | 136.6 cd           | 168.6 ab             | 160.3 <sup>b</sup> | 147.04             |
| Ni <sub>2</sub> | (5.12)               | (2.15)              | (4.89)               | (9.15)               | (1.14)             | (5.70)               | (3.46)             | 147.8 <sup>A</sup> |
| Main effects    | 74.96 <sup>°C</sup>  | 76.46 <sup>°C</sup> | 86.55 <sup>B</sup>   | 105.2 <sup>A</sup>   | 85.72 <sup>B</sup> | 104.3 <sup>A</sup>   | 106.9 <sup>A</sup> |                    |

**Notes:** Cl, control (without amendment addition); M, municipal solid waste compost; B, bentonite clay; R, municipal solid waste compost biochar; Ni<sub>0</sub>, without Ni application; Ni<sub>1</sub>, application of 150 mg Ni kg<sup>-1</sup> soil; Ni<sub>2</sub>, application of 300 mg Ni kg<sup>-1</sup> soil; WsEx-Ni, water and soluble form of Ni; Car-Ni, carbonate-bound Ni; Res-Ni, residual Ni.

Numbers with same lower letters (interaction effects), in each section, had no significance difference to each other (P < 0.05). Also, numbers with same capital letters (main effects), in each section, had no significance difference to each other (P < 0.05).

The interaction between Ni levels and soil amendments did not demonstrate statistically significant effects (P < 0.05) on altering Ni concentrations in the FeMnOx and OM fractions, although the main effects of each factor were individually significant (P < 0.05). Among the amendments, only B, R, and their combination (R+B) significantly reduced soil Ni content in the FeMnOx fraction compared to the control, whereas other treatments showed no significant impact. Nevertheless, no statistically significant differences were observed among the M, R, and R+B treatments (Figure 4). A plausible explanation is that over time, a portion of the soil Ni present in the FeMnOx fraction may undergo enhanced transformation into the more stable form like Res, particularly under biochar and zeolite amendments (Ali et al., 2019). The application of M and

M+R treatments significantly increased the Ni concentration in the OM fraction relative to the control, though no statistically significant difference was observed between these two treatments (Figure 4). The obtained result suggests that Ni ions preferentially form complexes with carboxyl functional groups on the M surface (as evidenced by C=O stretching vibrations at ~1700 cm<sup>-1</sup> in Figure 1), demonstrating stronger binding affinity compared to other passivation agents (Bashir et al., 2018). The application of biochar and compost amendments elevates soil organic carbon content, thereby improving its capacity for immobilizing both HMs and organic contaminants through enhanced adsorption mechanisms (Li et al., 2020; Puga et al., 2015). Previous studies have documented that biochar and compost amendments can significantly increase the proportion of soil HMs associated with the OM fraction (Li et al., 2021; Paradelo et al., 2018).

**Figure 4.** The main effects of amendments on changing the soil Ni concentration in the FeMnOx and OM fractions. Notes Cl, control (without amendment addition); M, municipal solid waste compost; B, bentonite clay; R, municipal solid waste compost biochar. Ni-FeMnOx, Ni bound to iron-manganese oxides; Ni-OM, Ni bound to organic matter. Numbers with same lower letters had no significance difference to each other (P < 0.05).

The Res fraction represents the most stable chemical form of HMs in soil, bound within the lattice structure of clay minerals and exhibiting no bioavailability (Shen et al., 2022). At the Ni<sub>0</sub> level, the application of all amendments significantly elevated the Ni concentration in the Res fraction compared to the control, with the greatest increase observed under the combined R+M treatment (3.5-fold) (Table 6). For the Ni<sub>1</sub> and Ni<sub>2</sub> levels, treatments including B, R, R+M, and R+B significantly enhanced the Ni content in the Res fraction relative to the control. Specifically, under Ni<sub>1</sub> conditions, the combined R+M treatment exhibited the highest Ni accumulation (+36.6%). In contrast, at the Ni<sub>2</sub> level, R alone produced the greatest increase (+29.6%), though their effects did not differ significantly from that of the R+B treatment (Table 6). The results suggest that as soil Ni contamination levels increase, the effectiveness of soil amendments in promoting Ni retention in the Res fraction appears to diminish (Table 6). Pearson correlation analysis revealed a significant positive relationship (r=0.42, P<0.01) between Ni concentration in the Res fraction and soil Olsen-P content. This correlation suggests that elevated phosphorus levels in the soil solution, resulting particularly from R and M amendments (which contain high P

concentrations as shown in Table 4), may enhance Ni presence in the Res fraction through the formation of insoluble Ni-phosphate compounds (Boostani et al., 2023). Biochar produced from biomass, particularly livestock manure and municipal solid waste enhance the bioavailability of P in soil upon application (Shi et al., 2023). The Res fraction, which incorporates metals within the crystal lattice of primary and secondary minerals (e.g., feldspars, micas, phosphates) and resistant minerals (zircon and rutile) (Shen et al., 2022), represents the most stable and geochemically inert pool. Its increase signifies a transition from bioavailable forms to a long-term sink, drastically reducing ecological risk through immobilization within a three-month incubation period (Boostani et al., 2024b).

412

416

428

435

The combined amendments did not generally perform better than the single amendments in altering soil Ni fractions. For example, at the Ni<sub>2</sub> level, M alone increased WsEx-Ni to 17.19 mg kg<sup>-1</sup>, whereas the combinations M+B (16.57 mg kg<sup>-1</sup>) and R+M (15.54 mg kg<sup>-1</sup>) did not surpass its effect. A similar pattern was observed for Car-bound Ni: M alone reached 32.09 mg kg<sup>-1</sup>, which was statistically higher than the combinations (M+B: 29.34 mg kg<sup>-1</sup>; R+M: 28.80 mg kg<sup>-1</sup>). Furthermore, the content of Ni in the Res fraction for R alone at Ni<sub>2</sub> level was 172.1 mg kg<sup>-1</sup>, again exceeding the levels achieved by the R+B (168.6 mg kg<sup>-1</sup>) and R+M (160.3 mg kg<sup>-1</sup>). These findings suggest that interactions among M, B, and R are not necessarily complementary. Possible reasons include competition for sorption sites or changes in soil pH and redox conditions that limit the cumulative effect. Overall, while combined amendments remain more effective than the untreated control, they did not demonstrate clear synergistic benefits. This implies that the added cost and complexity of using combinations may not be justified when a single, well-chosen amendment can achieve better results. In practice, it may be more efficient to select the amendment best suited to immobilizing the dominant Ni fraction in a particular soil. Finally, the sequential fractionation analysis of soil Ni revealed that all amendments except M promoted Ni immobilization by converting more accessible fractions (WsEx, Car, and FeMnOx) into the Res form. Notably, combined application of amendments including R+M, R+B ad M+B demonstrated no superior immobilization efficacy compared to individual treatments, with R emerging as the most effective amendment for Ni stabilization. Conversely, the M treatment appeared to enhance Ni mobility by increasing its concentration in bioavailable fractions (WsEx and Car). Mailakeba and Bk (2021) investigated the effects of incorporating kunai grass biochar at a rate of 0.75% into soils with varying Ni contamination levels (0, 56, 100, and 180 mg Ni kg<sup>-1</sup> soil). Their findings indicated that the biochar amendment enhanced Ni retention in the Res fraction while decreasing its presence in other soil fractions. Similarly, Boostani et al. (2023) reported that the addition of three different biochars derived from cow manure, municipal solid waste compost, and licorice root pulp, each applied at 3% (w/w), to a Ni-contaminated soil resulted in elevated Ni concentrations in the OM and Res fractions, alongside a reduction in Ni levels within the WsEx, Car, and FeMn oxide fractions. Table 7 summarized the main findings of our study alongside those reported in previous works on Ni fractionation and soil amendment effects. This allows a clearer contextualization of our results in the broader body of literature.

**Table 7.** Main findings of our study alongside those reported in previous works on Ni fractionation and soil amendment effects.

| Study                    | Amendments<br>Tested                                                                             | Main Findings on Ni Fractionation                                                                                                                                                                                                                                          | Consistency / Contrast<br>with Current Study                                                                                        |  |  |
|--------------------------|--------------------------------------------------------------------------------------------------|----------------------------------------------------------------------------------------------------------------------------------------------------------------------------------------------------------------------------------------------------------------------------|-------------------------------------------------------------------------------------------------------------------------------------|--|--|
| Present<br>study         | Biochar (R),<br>Bentonite (B),<br>Compost (M) and<br>their combinations<br>each at 2% (w/w)      | Increasing Ni levels (Ni <sub>0</sub> →Ni <sub>2</sub> ) increased Ni in FeMnOx (16.8-fold) and OM (15.4-fold). R most effective in reducing WsEx and increasing Res. M increased Ni in WsEx and Car fractions. Combined amendments not superior to individual treatments. | Confirms dominant<br>retention of Ni in<br>FeMnOx, OM, and Res<br>fractions. Highlights<br>biochar as most effective<br>stabilizer. |  |  |
| Boostani et al. (2020)   | crop residue<br>biochars (3% w/w)                                                                | Ni predominantly in Res, OM, and FeMnOx fractions. Biochar reduced Ni in Car fraction.                                                                                                                                                                                     | In agreement: both studies show Res, OM, FeMnOx as dominant sinks. Confirms biochar efficacy in shifting Ni into more stable pools. |  |  |
| Mailakeba &<br>Bk (2021) | Kunai grass biochar<br>(0–0.75%)                                                                 | Increasing Ni input (0 to 180 mg kg <sup>-1</sup> ) raised Ni in OM fraction; biochar enhanced Ni retention in Res fraction and reduced labile forms.                                                                                                                      | Aligns with our findings:<br>biochar improves Ni<br>stabilization and shifts Ni<br>from bioavailable to<br>stable forms.            |  |  |
| Gao & Li<br>(2022b)      | Bentonite (various doses)                                                                        | Dose-dependent reduction in Ni within Car fraction.                                                                                                                                                                                                                        | Consistent: B reduced Ni-Car in our study at moderate Ni (Ni <sub>1</sub> ).                                                        |  |  |
| Liang et al. (2021)      | Biochar (crop<br>residues)                                                                       | At high Ni loading, adsorption sites became saturated, reducing immobilization efficiency.                                                                                                                                                                                 | Supports our finding that R lost significant effectiveness at Ni <sub>2</sub> level due to site saturation.                         |  |  |
| Boostani et al. (2023)   | Biochars from cow<br>manure, municipal<br>solid waste<br>compost, licorice<br>root pulp (3% w/w) | Increased Ni in OM & Res fractions; decreased Ni in WsEx, Car, and FeMnOx.                                                                                                                                                                                                 | Matches to present study:<br>biochar promotes content<br>of Ni in the Res pool.                                                     |  |  |
| Bashir et al. (2018)     | compost                                                                                          | Ni formed strong complexes with carboxyl groups of compost.                                                                                                                                                                                                                | Explains our observation that M elevated Ni in the OM fraction.                                                                     |  |  |
| Ali et al. (2019)        | Biochar, zeolite                                                                                 | Promoted transformation of metals from FeMnOx fraction into more stable Res pool.                                                                                                                                                                                          | Agrees: our study found R and B enhanced Ni accumulation in Res fraction.                                                           |  |  |

# 3.6 Desorption pattern of soil Ni extracted by DTPA over time

The influence of various amendments on Ni release by DTPA solution over a 24-hour period is illustrated in Figure 5. Across all treatments, Ni desorption exhibited an initially rapid rate within the first two hours, followed by a slower release phase before reaching equilibrium by 24 hours (Figure 5). This biphasic pattern suggests that Ni desorption from the soil occurs in two distinct stages, each associated with different bonding energies (Boostani et al., 2019). The initial phase likely corresponds to more readily available Ni fractions, such as WsEx and Car forms, whereas the subsequent phase may be attributed to Ni associated with less labile fractions, including FeMnOx and OM pools (Boostani et al., 2023). Similar biphasic HMs release patterns have been frequently documented in previous studies (Jalali et al., 2019; Sajadi Tabar and Jalali, 2013; Taghdis et al., 2016).

Figure 5. Cumulative soil Ni release (mg kg<sup>-1</sup>) by DTPA solution over time for each treatment at different Ni levels.

Notes: Cl, control (without amendment addition); M, municipal solid waste compost; B, bentonite clay; R, municipal solid waste compost biochar; Ni<sub>0</sub>, without Ni application; Ni<sub>1</sub>, application of 150 mg Ni kg<sup>-1</sup> soil; Ni<sub>2</sub>, application of 300 mg Ni kg<sup>-1</sup> soil

Elevating Ni concentrations from Ni<sub>0</sub> to Ni<sub>2</sub> resulted in a substantial increase in Ni release. 469 470 Additionally, all amendments except M contributed to a reduction in soil Ni desorption (Figure 5). The highest cumulative Ni release over 24 hours was observed in the M×Ni<sub>2</sub> treatment, showing a 471 472 56.2% increase compared to the Cl×Ni<sub>2</sub> treatment. Conversely, the lowest Ni desorption at both Ni<sub>1</sub> and Ni<sub>2</sub> levels was recorded in the R treatment, with values of 12.84 mg Ni kg<sup>-1</sup> and 19.52 mg 473 Ni kg<sup>-1</sup>, respectively. These findings reaffirm that R was the most effective treatment for Ni 474 immobilization, whereas the sole application of M enhanced Ni mobility. However, the combined 475 476 application of M with B or R moderated and reduced Ni release compared to M alone at all the Ni levels (Figure 5). In a study, the minimal Cd release from a contaminated calcareous soil over a 477 478 24-hour period was recorded in the co-application treatment of municipal solid waste biochar and bentonite. This represented an 18.7% decrease in Cd desorption relative to the control (Boostani 479 et al., 2025). 480

# 3.7 Application of power function kinetics models to soil Ni desorption data

468

493

The effectiveness of various amendments in immobilizing Ni in soil was also evaluated using power function kinetic model. The release of Ni from soil, when extracted with DTPA solution over time, conformed to the power function kinetics model, demonstrating strong model fit with determination coefficients (R<sup>2</sup>) ranging from 0.86 to 0.99 and standard errors of estimate (SEE) between 0.08 and 0.18. These findings align with prior research establishing the power function equation as the most appropriate kinetic model for characterizing the desorption behavior of HMs in calcareous soils (Zahedifar and Moosavi, 2017; Sheikh-Abdullah et al., 2021; Boostani et al., 2025). The power function equation can be mathematically represented in two forms:  $q_t = at^b$ or in its linearized logarithmic form:  $Ln q_t = Ln a + b Ln t$ , where ' $q_t$ ' shows cumulative Ni desorbed at time t (mg Ni kg<sup>-1</sup> soil), 'a' indicates initial Ni desorption rate (mg Ni kg<sup>-1</sup> min<sup>-1</sup>), and 'b' represents desorption rate coefficient (mg Ni kg $^{-1}$ ) $^{-1}$ . The interaction effects of soil amendments and Ni rates on changing the magnitude of parameters derived from power function kinetic model were statistically significant (P <0.01). The kinetic parameters ( $\dot{a}$  and  $\dot{b}$ ) exhibit an inverse relationship with desorption behavior, where decreasing 'a' values and increasing 'b' values correspond to suppressed HMs release rates, as demonstrated by Dang et al. (1994). Among the single amendment treatments, R consistently demonstrated the lowest 'a' parameter values and highest 'b' parameter values across all Ni concentration levels (Table 8), indicating its superior efficacy in reducing Ni desorption from soil. Regarding combined amendments, the R+B treatment showed optimal performance in suppressing Ni release based on kinetic parameters, though statistical analysis revealed no significant difference between R+B and R alone (Table 8). The power function equation can be differentiated with respect to time (t) to obtain the instantaneous desorption rate:  $\frac{d_q}{d_t} = a.b.t^{b-1}$ , at t = 1, this expression reduces to:  $\frac{d_q}{d_t} = ab$ , where the product 'ab' corresponds to the initial desorption rate of the element from the soil matrix (Dalal, 1985). At the Ni<sub>0</sub> level, none of the applied amendments exhibited a statistically significant influence on the

'ab' parameter (Table 8). However, at the Ni<sub>1</sub> level, treatments containing M (M, M+B, and M+R) demonstrated a significant enhancement in 'ab' parameter values (Table 8). In contrast, under Ni<sub>2</sub> conditions, individual applications of B and R, as well as their combination (R+B), significantly reduced the 'ab' parameter values relative to the control. The most substantial reduction (30.3%) was observed for the R treatment (Table 8). A statistically significant and strongly positive correlation was observed between the 'ab' parameter and both the Ni-WsEx (r=0.90, P<0.01) and Ni-Car (r=0.97, P<0.01) fractions. These findings indicate that the initial release kinetics of Ni are governed by its concentration in the more labile fractions characterized by weaker adsorption energies.

**Table 8.** The Parameters of power function kinetics model as affected by the application of different amendments.

|                 | Cl                                              | M                   | В                       | R                     | M+B                  | R+B                  | R+M                | Main effects       |
|-----------------|-------------------------------------------------|---------------------|-------------------------|-----------------------|----------------------|----------------------|--------------------|--------------------|
|                 | a (mg Ni kg <sup>-1</sup> min <sup>-1</sup> ) b |                     |                         |                       |                      |                      |                    |                    |
| NT:             | 0.12 h                                          | $0.12^{h}$          | 0.11 h                  | 0.091 h               | 0.11 h               | 0.095 h              | 0.15 h             | 0.11 C             |
| Nio             | (0.007)                                         | (0.005)             | (0.014)                 | (0.007)               | (0.006)              | (0.004)              | (0.006)            | 0.11 <sup>C</sup>  |
| NI\$.           | 0.83 g                                          | 1.22 <sup>c-e</sup> | $0.90^{\text{ f-g}}$    | 0.74 <sup>g</sup>     | 1.27 cd              | $0.78^{g}$           | 1.15 d-f           | 0.98 <sup>B</sup>  |
| $Ni_1$          | (0.029)                                         | (0.016)             | (0.006)                 | (0.014)               | (0.019)              | (0.035)              | (0.033)            | 0.98               |
| Ni <sub>2</sub> | 1.47 bc                                         | 2.05 a              | 1.23 <sup>c-e</sup>     | $0.96^{\text{ e-g}}$  | 1.57 b               | 1.21 <sup>c-e</sup>  | 2.02 a             | 1.50 <sup>A</sup>  |
| 1N12            | (0.413)                                         | (0.029)             | (0.048)                 | (0.010)               | (0.013)              | (0.042)              | (0.007)            | 1.30               |
| Main effects    | $0.80^{\circ}$                                  | 1.13 <sup>A</sup>   | $0.74^{\circ}$          | 0.61 <sup>D</sup>     | 0.98 <sup>B</sup>    | 0.68 <sup>CD</sup>   | 1.10 AB            |                    |
|                 |                                                 |                     | b                       | (mg Ni kg-1)          | -1                   | _                    |                    |                    |
| NI:             | 0.18 g                                          | $0.19^{fg}$         | 0.21 ef                 | 0.23 e                | 0.23 e               | 0.23 e               | $0.18^{g}$         | 0.21 <sup>B</sup>  |
| $Ni_0$          | (0.016)                                         | (0.007)             | (0.021)                 | (0.011)               | (0.009)              | (0.005)              | ) (0.005)          | 0.21               |
| Niı             | 0.41 a-c                                        | 0.40 a-c            | 0.40 a-c                | 0.39 °                | 0.39 °               | $0.41^{\text{ a-c}}$ | 0.40 a-c           | 0.40 <sup>A</sup>  |
| INII            | (0.006)                                         | (0.001)             | (0.002)                 | (0.004)               | (0.006)              | (0.005)              | (0.011)            | 0.40               |
| $Ni_2$          | $0.40~^{\mathrm{a-c}}$                          | 0.38 °              | 0.41 a-c                | 0.42 a                | 0.42 a               | 0.40 a-c             | 0.36 <sup>d</sup>  | 0.40 <sup>A</sup>  |
| 1 <b>\1</b> 2   | (0.001)                                         | (0.001)             | (0.006)                 | (0.002)               | (0.001)              | (0.015)              | (0.008)            | 0.40               |
| Main effects    | 0.33 <sup>B</sup>                               | 0.33 <sup>B</sup>   | 0.34 <sup>A</sup>       | 0.35 <sup>A</sup>     | 0.35 <sup>A</sup>    | 0.35 <sup>A</sup>    | 0.32 <sup>B</sup>  |                    |
|                 |                                                 |                     |                         | ab                    |                      |                      |                    |                    |
| Nio             | 0.021 h                                         | 0.025 h             | $0.022^{h}$             | $0.021^{h}$           | 0.025 h              | 0.021 h              | 0.027 h            | 0.023 <sup>C</sup> |
| 1810            | (0.001)                                         | (0.001)             | (0.001)                 | (0.001)               | (0.001)              | (0.001)              | (0.001)            | 0.023              |
| Niı             | 0.338 g                                         | 0.491 de            | 0.365 fg                | $0.307^{\text{ g}}$   | $0.506^{\text{ de}}$ | $0.302^{g}$          | 0.457 ef           | 0.395 <sup>B</sup> |
| 1411            | (0.007)                                         | (0.006)             | (0.001)                 | (0.008)               | (0.001)              | (0.010)              | (0.002)            | 0.393              |
| $Ni_2$          | $0.587^{\text{ cd}}$                            | 0.797 <sup>a</sup>  | 0.503 de                | $0.409^{\text{ e-g}}$ | 0.663 bc             | $0.488^{\text{ de}}$ | 0.723 ab           | 0.595 <sup>A</sup> |
|                 | (0.164)                                         | (0.013)             | (0.012)                 | (0.003)               | (0.006)              | (0.001)              | (0.017)            | 0.333              |
| Main effects    | 0.315 <sup>B</sup>                              | 0.438 <sup>A</sup>  | $0.297  ^{\mathrm{BC}}$ | 0.245 <sup>C</sup>    | 0.398 <sup>A</sup>   | $0.270^{BC}$         | 0.402 <sup>A</sup> |                    |

**Notes:** Cl, control (without amendment addition); M, municipal solid waste compost; B, bentonite clay; R, municipal solid waste compost biochar; Ni<sub>0</sub>, without Ni application; Ni<sub>1</sub>, application of 150 mg Ni kg<sup>-1</sup> soil; Ni<sub>2</sub>, application of 300 mg Ni kg<sup>-1</sup> soil. Numbers with same lower letters (interaction effects), in each section, had no significance difference to each other (P < 0.05). Also, numbers with same capital letters (main effects), in each section, had no significance difference to each other (P < 0.05).

The use of an artificially contaminated soil was a necessary step to establish clear cause-effect relationships under controlled conditions, isolating the variables of interest (e.g., contaminant concentration and type of treatments) from the complex confounding factors present in the field. However, we acknowledge that historically contaminated field soils often exhibit reduced bioavailability and different sequestration patterns, which can influence remediation efficacy. It is also important to note that this study was conducted using a calcareous soil. Given that soil properties (e.g., texture, cation exchange capacity, organic matter and calcium carbonate content)

- are key determinants of desorption and adsorption process, the findings presented here may not be
- directly transferable to soils with significantly different characteristics. Future studies should be
- done to verify these results across a broader range of soil types.

## 4 Conclusions

- This study assessed the efficacy of R, M, and B, both individually and in combination, for
- immobilizing Ni in a contaminated calcareous soil. Sequential extraction revealed that all
- amendments except M successfully converted mobile Ni into a stable residual form. Contrary to
- expectations, combined treatments showed no synergistic effects, with R alone proving most
- effective. Desorption kinetics confirmed R's superior retention capacity, exhibiting the lowest Ni
- release. The lack of synergy in combined treatments provides crucial practical insight for
- policymakers and remediation projects, suggesting that simple, single-amendment strategies can
- be both effective and more economically viable. On the other hand, this research contributes a
- valuable, scalable solution for the in-situ remediation of HMs-contaminated soils, particularly in
- arid and semi-arid calcareous regions prevalent in many parts of the world. It is recommended that
- long-term trials coupled with advanced spectroscopic techniques (e.g., XAFS, XPS) to be done for
- confirmation the stability and speciation of Ni immobilized by these amendments. It is also
- suggested that Ni immobilization efficacy must be evaluated via using a broader range of
- historically contaminated soil types under plant cultivation. Furthermore, soil health parameters
- (microbial biomass, enzyme activities, nutrient availability) should be analyzed to confirm the
- remediation strategy does not impair soil fertility.
- Authors' Contributions H.R.B. Conceptualization, Formal analysis, Methodology, Investigation,
- Validation, Writhing the manuscript Z.J. Laboratory analyses A.B. Project administration, Review
- & Editing E.B. Review & Editing M. N. Methodology, Review & Editing.
- **Financial support.** No funding was received for conducting this study.
- Competing interests. The contact author has declared that neither they nor their co-authors have
- any competing interests.
- **Data availability.** The data generated in this study are available from the corresponding authors
- upon reasonable request.
- **Disclaimer.** Publisher's note: Copernicus Publications remains neutral with regard to
- jurisdictional claims in published maps and institutional affiliations.
- **Generative Artificial Intelligence (AI)** During the preparation of this work the author(s) used DeepSeek
- and Chat GPT (3.5) in order to improve the readability and language. After using this tool, the author(s)
- reviewed and edited the content as needed and take full responsibility for the content of the publication.
- **Acknowledgements.** This work was supported by College of Agriculture and Natural Resources of Darab,
- Shiraz University, Darab, Iran.

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
