# Peer review of "Comparative efficacy of individually and combined application of compost, biochar, and bentonite on Ni dynamics in a calcareous soil"

_EGUsphere, 2025_

## Author Comment (AC1)

**RESPONSE TO REVIEWER COMMENTS ON MANUSCRIPT: Comparative efficacy of individually and combined application of compost, biochar, and bentonite on Ni dynamics in a calcareous soil Egusphere-2025-2147**

The authors would like to thank anonymous reviewer for time, invaluable comments and suggestions for substantially improving this manuscript. Please find detailed responses to each comment below.

**ALL CHANGES ARE INDICATED IN GREEN HIGHLIGHT IN THE REVISED MANUSCRIPT**

**ANONYMOUS REFEREE #1**

**General comments**

The study titled "Com*arative efficacy of individually and combined application of compost, biochar, and bentonite on Ni dynamics in a calcareous soil"* evaluates the effectiveness of various soil amendments in reducing nickel (Ni) availability in a calcareous soil from Iran. Overall, the study is well-designed, and the results are clearly presented and thoughtfully discussed. While the topic is not entirely novel, it remains relevant, particularly in exploring the effectiveness of these specific amendments across different soil types and characteristics.

That said, the manuscript requires further revision before it can be considered for publication.

1.One of my main concerns is the selection of a calcareous soil, which is inherently high in pH. It is well established that pH is a major factor influencing metal availability in soils, and under alkaline conditions, metal solubility is generally very low. Therefore, the rationale behind selecting a soil type where Ni availability would not typically pose a significant problem should be clarified. Why was this specific soil chosen for the study?

**Authors' response:** We sincerely thank the reviewer for their insightful and valuable feedback regarding the selection of a calcareous soil for our study. The choice of a calcareous soil was driven by the specific objectives of our study, which aimed to investigate Ni dynamics in a context representative of agricultural soils of Iran where calcareous soils predominate. Calcareous soils, characterized by high calcium carbonate content and alkaline pH, are widespread in many arid and semi-arid climates, covering approximately 30% of global agricultural lands. Anthropogenic activities (e.g., industrial discharge, application of contaminated biosolids or fertilizers, mining) can lead to significant Ni accumulation in these soils. Although the immediate risk may be low, the long-term environmental fate and "latent" risk are critical assessment endpoints. Furthermore, Calcareous soils provide a rigorous testbed for an immobilizing amendment. If an amendment can further reduce the already low extractable pools of Ni or transform it into more stable phases (e.g., promoting formation of distinct Ni precipitates rather than just relying on adsorption at high pH), it demonstrates a robust and potentially superior remediation mechanism. This is a more challenging scenario than immobilizing metals in an acidic soil where simple pH elevation often has a dramatic effect. Success here suggests the amendment could be effective across a wider range of pH conditions. To address the reviewer's concern, we propose to enhance the clarity of our rationale in the revised manuscript by adding the following text to the Materials and Methods section (2.1.):

//The selection of a calcareous soil for this study was motivated by its prevalence in agricultural soils of Iran and the need to address a knowledge gap regarding Ni dynamics in high-pH environments. Contrary to the well-established principle that high pH reduces metal solubility, recent studies suggest that specific

pedogenic factors and anthropogenic contamination may facilitate increased Ni mobilization in calcareous soils (Shahbazi et al., 2022)//

2.Additionally, to better contextualize the work, it would be helpful to include the threshold values for Ni contamination in Iranian soils. This information would provide a clearer understanding of the extent to which the applied treatments are potentially effective or relevant.

**Authors' response**: We thank the referee for this insightful and constructive comment. We have added the threshold value for Ni contamination in Iranian soils within the introduction section. Please see the revised manuscript.

The threshold value for Ni contamination in Iranian soils is 110 mg kg$^{-1}$ soil (Shahbazi et al., 2020).

3.Regarding the non-contaminated soil used in the study, I believe it is not appropriate to assess the effectiveness of the amendments in such a context. When Ni concentrations are already very low, the changes induced by treatments may be below quantifiable limits, and the results could be unreliable. I recommend removing these results from the text. Please include the quantification and detection limits for Ni used in the analytical methods section.

**Authors' response**: The detection limit for Ni quantification using a PG990 AAS in flame mode (Air/Acetylene) is 3.0 μg L$^{-1}$ (ppb) under optimized analytical conditions. In the non-contaminated soil sample, the total Ni concentration was measured at 38 mg kg$^{-1}$, while the WsEx fraction contained 6.75 mg kg$^{-1}$. Consequently, the concentrations of Ni across all measured chemical fractions were found to be orders of magnitude above the instrumental detection limit, thereby confirming the suitability of the PG990 AAS for reliable quantification in this study. We have added the detection limits for Ni used in the analytical methods section.

2.8.4 *Detection limits and precision*

The detection limit for Ni with the PG990 in flame mode was ~3 μg L$^{-1}$ under optimized conditions. Analytical precision was verified by duplicate sample runs, with relative standard deviations (RSDs) consistently below 5%.

**Specific comments:**

4.In the Materials and Methods section (2.5), please provide detailed steps and reagents used for the sequential extraction procedure. Given that the residual fraction is frequently discussed in the manuscript as a potential sink for metal removal, it would be valuable to elaborate on its potential composition and significance. In particular, please address the plausibility of metals being incorporated into mineral phases (which are the typically implied when referring to the residual phase) within a two-month period.

**Authors' response**: thanks for your excellent suggestion. We have added a Table in the Materials and Methods section (2.5) for providing detailed steps and reagents used for the sequential extraction procedure. We have also expanded the Discussion to include a more detailed examination of the composition and environmental significance of the Residual fraction, as now detailed in the revised manuscript.

**Table 2.** Sequential extraction procedure (Salbu and Krekling, 1998).

| Fractions | Reagent | Agitation time |
|---|---|---|
| Soluble+exchangeable (WsEx) | 20 mL of 1 M NH$_4$OAc (pH=7) | 2 h |
| Carbonate-bound (Car) | 20 mL of 1 M NH$_4$OAc (pH=5) | 2 h |

| | | |
|---|---|---|
| FeMnOxide-bound (FeMnOx) | 20 mL of 0.04 M $NH_2OH \cdot HCl$ in 25 % (v/v) $CH_3COOH$ | 6 h at 60°C |
| Organic-bound (OM) | 15 mL of 30 % (v/v) $H_2O_2$ (pH=2, $HNO_3$) | 5.5 h at 80°C (on water bath) |
| | and after cooling, 5 mL of 3.2 M $NH_4OAc$ in 20 % (v/v) $HNO_3$ | 30 min |
| Residual-bound (Res) | Calculated by subtracting the sum of four fractions from total Pb | |

The Res fraction, which incorporates metals within the crystal lattice of primary and secondary minerals (e.g., feldspars, micas, phosphates) and resistant minerals (zircon and rutile) (Shen et al., 2022), represents the most stable and geochemically inert pool. Its increase signifies a transition from bioavailable forms to a long-term sink, drastically reducing ecological risk through immobilization within a three-month incubation period (Boostani et al., 2024b).

5.Lines 385–387: The metal-phosphate fraction should not be classified as part of the residual fraction. Please revise this categorization.

**Authors' response**: We thank the reviewer for this valuable comment. In many schemes, Ni–phosphate precipitates are classified as a specific fraction (e.g., metal-phosphate) or included in the residual fraction if tightly bound in mineral lattices. In the present study, there was a positive and significant correlation between residual fraction and Olsen-P. Due to this, we prefer to classify it as residual fraction.

6.Lines 351–352 and 358–359: Biochar contributes carbon, but not organic matter per se. This distinction should be corrected.

**Authors' response:** We thank the reviewer for this insightful comment. The referee is correct to highlight the important technical distinction between carbon and organic matter. We have corrected the manuscript to clarify. Please see the revised manuscript.

7.To avoid confusion, consider changing the abbreviation "B" used for either biochar or bentonite in the treatment labels. Using the same letter for both may lead to misinterpretation.

**Authors' response:** We thank the referee for this helpful suggestion. We have revised the manuscript to use 'R' for biochar and 'B' for bentonite in all treatment labels and text to avoid any potential confusion. Please see the revised manuscript.

8.In all tables, there appears to be an extra row and an extra column without titles. Please clarify their meaning or remove them if they are not necessary. In addition, it is unclear what the comparisons indicated by capital letters and lowercase letters represent.

**Authors' response:** We sincerely thank the referee for their careful review and for identifying this lack of clarifications. We have now revised all tables in the manuscript to address both points. Please see the revised manuscript.

10.Include standard deviation in tables and graphs

**Authors' response**: We thank the referee for this important comment. We have now included the standard deviation values for all data in Tables (3, 4, 5, 6, 7). Furthermore, error bars representing the standard deviation have been added to all relevant figures (Figures 4). These changes provide a clear indication of data variability throughout the manuscript. Please see the revised manuscript.

11.Figure 5 requires revision to enhance clarity and interpretability. I recommend applying a consistent color palette or uniform symbols across treatments or Ni doses to facilitate direct comparison. Additionally, it would be beneficial to present, within the same figure, a series of smaller panels depicting the kinetics by treatment group (or by Ni level group)

Authors' response: We thank the referee for this excellent suggestion. To enhance interpretability, Figure 5 was reconfigured into a series of panels grouped by treatment, each showing the kinetic data for all Ni doses. Uniform axis scales were maintained across all panels to ensure comparability. Please see the revised manuscript.

12.Finally, please include some discussion about the fact that these results are obtained from an artificially contaminated soil, and those could change with a field contaminated soil.

**Authors' response:** We thank the reviewer for this important comment. We fully agree that results from artificially contaminated soil can differ from those in field-contaminated soils, and we have now included a concise discussion on this point within the revised manuscript (and conclusion sections).

The End of discussion….

//The use of an artificially contaminated soil was a necessary step to establish clear cause-effect relationships under controlled conditions, isolating the variables of interest (e.g., contaminant concentration and type of treatments) from the complex confounding factors present in the field. However, we acknowledge that historically contaminated field soils often exhibit reduced bioavailability and different sequestration patterns, which can influence remediation efficacy//.

The End of conclusion…

// It is also suggested that Ni immobilization efficacy by these amendments must be evaluated via using a broader range of historically contaminated soil types under plant cultivation //

---

## Author Comment (AC2)

**RESPONSE TO REVIEWER COMMENTS ON MANUSCRIPT: Comparative efficacy of individually and combined application of compost, biochar, and bentonite on Ni dynamics in a calcareous soil Egusphere-2025-2147**

The authors would like to thank the reviewer for time, invaluable comments and suggestions for substantially improving this manuscript. Please find detailed responses to each comments below.

**ALL CHANGES ARE INDICATED IN GREEN HIGHLIGHT IN THE REVISED MANUSCRIPT**

**Referee #2: Abhishek Kumar, abikumar@ucdavis.edu**

The authors investigated an important issue of Ni contamination in calcareous soils and evaluated compost, biochar, and bentonite for Ni immobilization. While the study addresses a relevant topic, the manuscript shows weaknesses in depth of analysis, clarity of presentation, and use of English. Substantial revision is needed before further consideration. Key areas for improvement are outlined below:

**Abstract**

1.The abstract should be more concise and focused, emphasizing the novelty, main findings, and broader implications. The knowledge gap is vaguely stated. Please clarify how this study differs from earlier works.

Authors' response: We sincerely thank the reviewer for thoughtful and constructive comments, which have significantly helped us improve the clarity and impact of the abstract. We have carefully considered all points raised and have revised the abstract accordingly. The new abstract is as follows: In Iran, Ni-contaminated calcareous soils pose a significant environmental risk, yet effective remediation strategies for these specific conditions remain underexplored. While organic and inorganic amendments are commonly used, their comparative efficacy and potential synergistic effects in combined applications for Ni immobilization are not well-established. To address this, an incubation study investigated the individual and combined effects of municipal solid waste compost (M), its biochar (R), and bentonite (B) on Ni stabilization in a calcareous soil at three Ni-contamination levels (0, 150, and 300 mg Ni kg$^{-1}$). Sequential extraction and DTPA-release kinetics demonstrated that R was the most effective treatment, significantly reducing labile Ni by transforming it into the residual fraction. This is likely due to its alkaline pH, ash, and phosphorus content, which promote Ni precipitation. In contrast, M increased soil Ni bioavailability. The results revealed that combinations (M+B, R+B, R+M) offered no synergistic advantage. The main finding was that singly-applied municipal solid waste biochar is a superior amendment for Ni immobilization, providing a more efficient and practical remediation strategy for contaminated calcareous soils without the need for complex combined treatments.

**Introduction**

2.The novelty of the study is not highlighted clearly or justified adequately.

Authors' response: We thank the referee for this critical feedback. We agree that the original statement was too vague. We have now revised the text to explicitly define the specific knowledge gap our study addresses and to clearly articulate the novel aspect of our work. The new statement is as follows: While previous studies have examined individual organic and inorganic amendments for metal immobilization, a critical knowledge gap exists regarding the comparative and synergistic efficacy of compost, biochar, and bentonite on Ni immobilization in contaminated calcareous soils. It remains uncertain whether combined applications offer a superior strategy over single amendments for immobilizing Ni in calcareous environments, a key question this study aimed to answer.

3.The hypotheses are mentioned but not logically developed from the background information.

**Authors' response:** We thank the referee for this valuable comment. We have now revised them to explicitly state the logical reasoning and scientific principles upon which they are based, ensuring a stronger and more justified connection to the background of the study. The hypotheses were changed as follows: Based on the well-documented properties of high stability, alkaline pH, and significant surface area in biochars, we hypothesized that biochar would be more effective than compost or bentonite at immobilizing Ni, by converting more labile fractions into the stable residual form. Furthermore, given the distinct mechanisms of metal retention offered by organic (e.g., complexation) and inorganic (e.g., adsorption, ion exchange) amendments, we also hypothesized that their combined application would exhibit a synergistic effect, leading to a greater reduction in Ni bioavailability and desorption than any amendment applied alone.

4.The objectives should be clearly stated in a structured format (e.g., i, ii, iii or a, b, c).

**Authors' response:** We thank the referee for this suggestion. We have revised the objectives section to present them in a structured, enumerated list as recommended. The objectives have also been refined to be more specific and measurable. The revised sentence is as follows: Therefore, the goals of the present study were to (i) Compare the effectiveness of municipal solid waste compost, its derived biochar, and bentonite, both individually and in combination, on immobilizing Ni in an artificially contaminated calcareous soil by assessing changes in its chemical fractions, (ii) Determine the kinetics of Ni release in the amended soils to evaluate the stability of immobilized Ni, and (iii) Identify the primary mechanisms responsible for Ni immobilization by employing advanced analytical techniques.

**Methodology**

5.Provide a rationale or reference for selecting the 2% w/w amendment rate.

**Authors' response:** Thanks for your valuable suggestion. We have added an international reliable reference for selecting the 2% w/w amendment rate. Please see the revised manuscript.

W. Jiang, Y. Liu, J. Zhou, H. Tang, G. Meng, X. Tang, et al. 2024. Biochar co-compost increases the productivity of Brassica napus by improving antioxidant activities and soil health and reducing lead uptake. Frontiers in Plant Science, 15, 1475510.

6.Only one soil type was studied, which limits the generalizability of the findings. This limitation should be acknowledged.

**Authors' response:** Thank you for this insightful comment. We have now explicitly acknowledged this limitation in the revised manuscript in the conclusion section. The new sentence is as follows: It is also suggested that Ni immobilization efficacy by these amendments must be evaluated via using a broader range of historically contaminated soil types under plant cultivation. Specifically, in the fourth paragraph of the Discussion, we now state: It is also important to note that this study was conducted using a calcareous soil. Given that soil properties (e.g., texture, cation exchange capacity, organic matter and calcium carbonate content) are key determinants of desorption and adsorption process, the findings presented here may not be directly transferable to soils with significantly different characteristics. Future studies should be done to verify these results across a broader range of soil types. Please see the revised manuscript.

7.Include information on quality assurance/quality control (QA/QC) procedures.

**Authors' response:** We appreciate the reviewer's comment regarding the inclusion of quality assurance/quality control (QA/QC) procedures. In the revised manuscript, we have added the following

details to the *Materials and Methods* section to clarify the steps undertaken to ensure data reliability and reproducibility. Please see the revised manuscript.

2.8 Quality assurance/quality control (QA/QC) procedures

2.8.1 *Soil and amendment preparation*

All soils and amendments (R, M, B) were processed using clean, acid-washed tools to prevent cross-contamination. Samples were homogenized by thorough mixing before subsampling for analyses.

2.8.2 *Replicates and blanks*

All treatments were performed in triplicate, and reagent blanks were included during sequential extraction and desorption procedures to monitor potential contamination.

2.8.3 *Calibration and standards*

Nickel concentrations were quantified using AAS (PG 990, PG Instruments Ltd., UK). The instrument was calibrated daily with a series of standard Ni solutions prepared from certified stock solutions (Merck, Germany). A calibration curve with $r^2 > 0.999$ was obtained before each run. Quality control standards were analyzed after every 10 samples to confirm instrument stability.

2.8.4 *Detection limits and precision*

The detection limit for Ni with the PG990 in flame mode was ~3 µg L$^{-1}$ under optimized conditions. Analytical precision was verified by duplicate sample runs, with relative standard deviations (RSDs) consistently below 5%.

2.8.5 *Sequential extraction*

To minimize redistribution or re-adsorption of Ni during sequential extraction, all extractions were performed in acid-cleaned polypropylene centrifuge tubes with constant agitation under controlled conditions.

2.8.6 *Statistical validation*

Normality and homoscedasticity were confirmed to validate the assumptions for ANOVA and Tukey's test.

**Results and Discussion**

8.Abbreviations should be defined only at first use, not repeated in every section.

**Authors' response:** Thank you for this helpful comment. We apologize for the oversight. We have now carefully reviewed the entire manuscript to ensure that each abbreviation is defined only upon its first use in the main text, abstract, and figure/table legends. Please see the revised manuscript.

9.The results of "combined amendments" are under-discussed. The absence of synergistic effects should be analyzed more critically.

**Authors' response**: We appreciate the referee's observation regarding the limited discussion of combined amendments and the absence of synergistic effects. We have added a discussion for showing the absence of synergistic effects in combined treatments (M+B, R+B, R+M). please see the revised manuscript.

The combined amendments did not generally perform better than the single amendments in altering soil Ni fractions. For example, at the $Ni_2$ level, M alone increased WsEx-Ni to 17.19 mg kg$^{-1}$, whereas the combinations M+B (16.57 mg kg$^{-1}$) and R+M (15.54 mg kg$^{-1}$) did not surpass its effect. A similar pattern was observed for Car-bound Ni: M alone reached 32.09 mg kg$^{-1}$, which was statistically higher than the combinations (M+B: 29.34 mg kg$^{-1}$; R+M: 28.80 mg kg$^{-1}$). Furthermore, the content of Ni in the Res fraction for R alone at $Ni_2$ level was 172.1 mg kg$^{-1}$, again exceeding the levels achieved by the R+B (168.6 mg kg$^{-1}$) and R+M (160.3 mg kg$^{-1}$). These findings suggest that interactions among M, B, and R are not necessarily complementary. Possible reasons include competition for sorption sites or changes in soil pH and redox conditions that limit the cumulative effect. Overall, while combined amendments remain more effective than the untreated control, they did not demonstrate clear synergistic benefits. This implies that the added cost and complexity of using combinations may not be justified when a single, well-chosen amendment can achieve better results. In practice, it may be more efficient to select the amendment best suited to immobilizing the dominant Ni fraction in a particular soil.

10. Study limitations—such as the use of a single soil, reliance on incubation rather than field experiments, and the lack of plant uptake validation—should be explicitly acknowledged.

Authors' response: Thank you for these valuable suggestions. We have considered all of them in the revised manuscript in the end of discussion and conclusion sections. Please see the revised manuscript.

11.The manuscript shows formatting issues (justified alignment with excessive spacing); this should be corrected.

**Authors' response:** We thank the reviewer for pointing this out. The manuscript has now been reformatted to eliminate excessive spacing. The text now has a uniform and clean appearance throughout. Please see the revised manuscript.

12.Consider adding a comparative table summarizing results from this study alongside findings from previous studies.

**Authors' response**: We thank the reviewer for the valuable suggestion. We have included a comparative table summarizing the main findings of our study alongside those reported in previous works on Ni fractionation and soil amendment effects. Please see the revised manuscript.

| Study | Amendments Tested | Main Findings on Ni Fractionation | Consistency / Contrast with Current Study |
|---|---|---|---|
| Present study | Biochar (R), Bentonite (B), Compost (M) and their combinations each at 2% (w/w) | Increasing Ni levels ($Ni_0 \rightarrow Ni_2$) increased Ni in FeMnOx (16.8-fold) and OM (15.4-fold). R most effective in reducing WsEx and increasing Res. M increased Ni in WsEx and Car fractions. Combined amendments not superior to individual treatments. | Confirms dominant retention of Ni in FeMnOx, OM, and Res fractions. Highlights biochar as most effective stabilizer. |

Table 7. Main findings of our study alongside those reported in previous works on Ni fractionation and soil amendment effects.

| Boostani et al. (2020) | crop residue biochars (3% w/w) | Ni predominantly in Res, OM, and FeMnOx fractions. Biochar reduced Ni in Car fraction. | In agreement: both studies show Res, OM, FeMnOx as dominant sinks. Confirms biochar efficacy in shifting Ni into more stable pools. |
|---|---|---|---|
| Mailakeba & Bk (2021) | Kunai grass biochar (0–0.75%) | Increasing Ni input (0 to 180 mg kg$^{-1}$) raised Ni in OM fraction; biochar enhanced Ni retention in Res fraction and reduced labile forms. | Aligns with our findings: biochar improves Ni stabilization and shifts Ni from bioavailable to stable forms. |
| Gao & Li (2022b) | Bentonite (various doses) | Dose-dependent reduction in Ni within Car fraction. | Consistent: B reduced Ni-Car in our study at moderate Ni (Ni$_1$). |
| Liang et al. (2021) | Biochar (crop residues) | At high Ni loading, adsorption sites became saturated, reducing immobilization efficiency. | Supports our finding that R lost significant effectiveness at Ni$_2$ level due to site saturation. |
| Boostani et al. (2023) | Biochars from cow manure, municipal solid waste compost, licorice root pulp (3% w/w) | Increased Ni in OM & Res fractions; decreased Ni in WsEx, Car, and FeMnOx. | Matches to present study: biochar promotes content of Ni in the Res pool. |
| Bashir et al. (2018) | compost | Ni formed strong complexes with carboxyl groups of compost. | Explains our observation that M elevated Ni in the OM fraction. |
| Ali et al. (2019) | Biochar, zeolite | Promoted transformation of metals from FeMnOx fraction into more stable Res pool. | Agrees: our study found R and B enhanced Ni accumulation in Res fraction. |

**Conclusions**

13.Beyond summarizing findings, the conclusions should place the study in a broader global context.

**Authors' response:** Thanks for your constrictive comment. We have summarized the findings in conclusion section and provided the conclusion in a broader global context with certain suggestions for future works. The new conclusion is as follows: This study assessed the efficacy of biochar (R), compost (M), and bentonite (B), both individually and in combination, for immobilizing Ni in a contaminated calcareous soil. Sequential extraction revealed that all amendments except M successfully converted mobile Ni into a stable residual form. Contrary to expectations, combined treatments showed no synergistic effects, with R alone proving most effective. Desorption kinetics confirmed R's superior retention capacity, exhibiting the lowest Ni release. The lack of synergy in combined treatments provides crucial practical insight for policymakers and remediation projects, suggesting that simple, single-amendment strategies can be both effective and more economically viable. On the other hand, this research contributes a valuable, scalable solution for the in-situ remediation of HMs-contaminated soils, particularly in arid and semi-arid calcareous regions prevalent in many parts of the world. It is recommended that long-term trials coupled with advanced spectroscopic techniques (e.g., XAFS, XPS) to be done for confirmation the stability and speciation of Ni immobilized by these amendments. It is also suggested that Ni immobilization efficacy by these amendments must be evaluated via using a broader range of historically contaminated soil types under plant cultivation. Furthermore, soil health parameters (microbial biomass, enzyme activities, nutrient

availability) should be analyzed to confirm the remediation strategy does not impair soil fertility. Please see the revised manuscript.

14.Limitations encountered during the study should be clearly noted.

**Authors' response**: Thanks for the valuable comment. We have considered all the limitations including the use of a single soil, reliance on incubation rather than field experiments, and the lack of plant uptake validation in the conclusion section. These limitations were indicated as follows: It is also suggested that Ni immobilization efficacy by these amendments must be evaluated via using a broader range of historically contaminated soil types under plant cultivation. Please see the revised manuscript.

15.Provide at least three concrete suggestions for future research directions.

**Authors' response:**  Thanks for your valuable suggestion. We have provided three concrete suggestions for future research as follows: It is recommended that long-term trials coupled with advanced spectroscopic techniques (e.g., XAFS, XPS) to be done for confirmation the stability and speciation of Ni immobilized by these amendments. It is also suggested that Ni immobilization efficacy by these amendments must be evaluated via using a broader range of historically contaminated soil types under plant cultivation. Furthermore, soil health parameters (microbial biomass, enzyme activities, nutrient availability) should be analyzed to confirm the remediation strategy does not impair soil fertility.

---

## Author Response (AR2)

RESPONSE TO REVIEWER COMMENTS ON MANUSCRIPT: Comparative efficacy of individually and combined application of compost, biochar, and bentonite on Ni dynamics in a calcareous soil Egusphere-2025-2147

The authors would like to thank anonymous reviewer for time, invaluable comments and suggestions for substantially improving this manuscript. Please find detailed responses to each comment below.

ALL CHANGES ARE INDICATED IN YELLOW IN THE REVISED MANUSCRIPT

**REFEREE #1**

I have only a minor comment that should be clarified by the authors before publication. When referring to the detection limit in  $\mu g \ L^{-1}$ , this value applies to the solution. However, when discussing Ni concentration in soil, you are converting this concentration from the solution, which depends (for example) on the amount of soil used in your Ni extraction protocol. Therefore, the detection limit of the AAS cannot be directly compared with the Ni concentration in soil. In summary, please ensure that all references to detection limits are made with respect to your Ni extraction solutions.

Authors' response: In response to this valuable feedback, we have revised the manuscript (2.8.4 *Detection limits and precision*) to ensure all references to detection limits are explicitly and correctly contextualized. The changes are as follows:

"Based on this detection limit and our extraction protocols, which used a Y g soil sample diluted to a final volume of Z mL, the corresponding method detection limit for Ni in soil was calculated as mg kg-1)"

In addition, we have added DOI number for each reference wherever possible.